# Simultaneous EEG-fMRI: What Have We Learned and What Does the Future Hold?

**DOI:** 10.3390/s22062262

**Published:** 2022-03-15

**Authors:** Tracy Warbrick

**Affiliations:** Brain Products GmbH, Zeppelinstrasse 7, 82205 Gilching, Germany; tracy.warbrick@brainproducts.com

**Keywords:** EEG, fMRI, simultaneous EEG-fMRI, multimodal imaging, brain imaging, cognitive neuroscience

## Abstract

Simultaneous EEG-fMRI has developed into a mature measurement technique in the past 25 years. During this time considerable technical and analytical advances have been made, enabling valuable scientific contributions to a range of research fields. This review will begin with an introduction to the measurement principles involved in EEG and fMRI and the advantages of combining these methods. The challenges faced when combining the two techniques will then be considered. An overview of the leading application fields where EEG-fMRI has made a significant contribution to the scientific literature and emerging applications in EEG-fMRI research trends is then presented.

## 1. Introduction

Understanding the structure and function of the human brain is one of the most fascinating, yet challenging, of scientific endeavours. The brain is the centre for coordinating the senses, regulating emotions, processing our thoughts, and initiating actions. Understanding its function in both healthy and diseased states can provide a valuable insight into human behaviour. As such, the brain has been the focus of scientific interest for many years leading to remarkable developments in technology that enable the measurement of different aspects of brain structure and function. The development of methods such as electroencephalography (EEG), magnetoencephalography (MEG), magnetic resonance imaging (MRI), positron emission tomography (PET), and functional near-infrared spectroscopy (fNIRS) has pushed the fields of brain imaging and cognitive neuroscience to grow into dynamic and progressive research disciplines. As new technologies develop into routine techniques, a natural progression is to begin to combine the different technologies to further our understanding beyond what we can learn from a single modality alone. Multimodal brain imaging has seen a surge in recent years [1] with simultaneous EEG-fMRI being at the forefront of cognitive multimodal neuroimaging. The significance of EEG-fMRI for multimodal brain imaging can be seen in the steady increase of papers in the first years of its application to the plateau in recent years as the technique has become established (Figure 1). This review paper covers the development of EEG-fMRI into a mature cognitive neuroscience technique and considers what contributions to the literature have been possible because of this multimodal imaging method over the past 25 years. It will focus on human EEG-fMRI except where animal research is related and focused on scalp EEG rather than invasive EEG unless it is relevant to the specific topic being considered.

First, the measurement principles involved in EEG and fMRI will be introduced and then why combining these techniques is advantageous will be considered. Next, the challenges faced when combining the two techniques and how the scientific community has met those challenges will be covered. The main part of the paper considers the contribution of EEG-fMRI to different application fields and the cutting-edge EEG-fMRI research and trends that hint to future developments. Note that this is not a comprehensive review of all EEG-fMRI papers but rather an overview of the main areas in which EEG-fMRI has made a significant difference, how EEG-fMRI has shaped technological developments, and applications that have brought or will bring us a new insight to brain function.

## 2. Measurement Principles

The motivation for combining EEG and fMRI measurements is due to the complementary properties of the signals measured in each modality and what the combined measurement of those signals can tell us about brain function.

### 2.1. EEG

When our sensory system is presented with a stimulus or a change in state a population of neurons fire. An area of synchronised neural activity results in a measurable potential at the scalp and this is what we measure using EEG. EEG electrodes measure the electric potential differences (voltages) that are generated by the synchronised neural activity, but this electrical signal must pass through the brain and skull and due to volume conduction, they reach the electrode in an attenuated form [2]. As such, scalp EEG can be considered an indirect measure of neural activity. The volume conduction problem also means that it is difficult to infer from where the electrical potentials measured on the scalp originate, this makes EEG source localization an ill-posed problem. The sources of neural activity are in the 3-dimensional brain while EEG only measures the 2-dimensional surface. The signals measured at the scalp surface do not represent the location of the active neurons generating the signal [3]; however, EEG measures neural activity on a millisecond timescale allowing us to access the rapidly changing dynamics of neuronal populations. This has made EEG a valuable research tool for understanding brain function for almost a century [4].

### 2.2. fMRI

An increase in neural activity stimulates higher energy consumption and increased blood flow, and this mechanism is known as neurovascular coupling. This increase in metabolic demand results in an increased blood oxygen level in the activated brain region. We can detect this difference in the blood oxygen level by taking advantage of the magnetic characteristics of haemoglobin; oxygenated blood has a different magnetic signature than de-oxygenated blood [5]. As such, fMRI uses the blood oxygenation level dependent (BOLD) imaging technique. BOLD fMRI can be considered a correlate of neural activity rather than a measure of neural activity since it represents the complex process of neurovascular coupling and the interaction of circulatory and metabolic demands [6]. These changes in blood flow occur seconds after neural activity changes, much more slowly than the millisecond timescale on which we measure EEG. Therefore, the temporal resolution of BOLD fMRI is limited by the slow haemodynamic response [7] so it does not easily provide information on when these activations occur; however, fMRI provides a 3-dimensional map of regional brain activity and a sub-millimetre spatial resolution can be achieved [6], thus allowing the spatial localisation of brain activity that we cannot achieve with EEG.

### 2.3. Combining EEG and fMRI

Given the strengths and deficits of each method, combining them has the potential to provide insight into brain function that cannot be measured by one modality on its own. Obtaining complementary datasets in response to the same changes in spontaneous or evoked brain activity is an attractive prospect. This is particularly valuable in scenarios where separate sessions would not be able to capture the same activity, for example spontaneous resting state fluctuations, responses to a task in a cognitive neuroscience experiment, or endogenous activity such as epileptic spikes. Moreover, there are some other small but notable advantages of simultaneous EEG-fMRI over single sessions, for example, eliminating habituation effects, having the same sensory environment for all measures (sitting in a lab versus lying in a noisy scanner), and practical aspects such as reducing the overall time required for the experiment [8]. However, it should be noted that simultaneous EEG-fMRI is not always necessary or appropriate; the availability of a technique does not mean it should always be used. Combing the two techniques comes with certain disadvantages, particularly regarding data quality (see Section 3.2). Careful consideration of whether simultaneous EEG-fMRI is the right tool for answering the research question is required. For a review and guidance on whether simultaneous EEG-fMRI is an appropriate method for a study see Scrivener (2020) [9].

## 3. Challenges

Multimodal imaging is not without its challenges and while it offers many advantages it also requires compromises. The need for compromise is often overlooked in the enthusiasm for exploiting the potential advantages but it must be considered if the technique is to be implemented successfully. The MR environment presents some significant difficulties for safely recording meaningful EEG. In turn, the magnetic fields of the MRI scanner can be disrupted by the presence of the EEG system inside the scanner. These interactions have implications for the safety of simultaneous EEG-fMRI recordings and the quality of both the EEG and fMRI data. The goal of a successful multimodal imaging setup is to reduce the amount of compromise necessary so the data obtained from both measures can be optimised and measured safely. Such challenges motivate the research community and industry to look for better solutions so that the technique can develop into a mature accepted neuroimaging method. The desire to combine EEG and fMRI to answer questions about brain function has driven the technological development of hardware, software, methods for data processing to retrieve clean data from both modalities, and methods for integrating the two datasets. In this section the challenges faced by the early pioneers of simultaneous EEG-fMRI will be considered and the technological and processing methods that were developed to allow this technique to progress will be discussed.

### 3.1. Safety

By necessity the EEG system contains electrically conductive materials, and it requires that a conductive material is placed near the participant’s head [10]. It is also worth noting that the EEG system is a recording circuit (EEG electrode versus reference electrode) and consists of electrodes, lead wires, an amplifier, and a power source. For simultaneous EEG-fMRI measurements this system is placed inside a static magnetic field (B0), and during scanning strong switching gradient fields and radio frequency (RF) fields (B1) are used. The main concern related to the static magnetic field is displacement force and torque. The primary safety concern related to the electrical conductivity of the EEG system is heating. The RF fields and switching gradient fields induce electromotive forces which generate a voltage that causes a current in the conductive loop formed by the electrodes and leads wires, and this can result in heating of the components and the local tissue [11]. In addition, antenna effects where a current can be induced along the length of a lead wire are also possible. RF-coupling is the dominant source of induced heating [10] but eddy currents induced by the gradient system can also result in the heating of metallic components, particularly if a high duty cycle is employed [12]. RF related heating is a little more complicated than gradient induced eddy currents because resonance effects play a role in RF heating, and these can be difficult to predict accurately because different configurations of the equipment might lead to different heating patterns. In addition, the effects of the EEG system on the B1 field can lead to changes in the B1 distribution across the head which results in unpredictable changes to local heating [10,13]. These factors need to be considered when developing systems suitable for EEG-fMRI. Here, an overview of safety related considerations is presented; for a comprehensive review of safety related issues for both scalp and intracranial EEG please see Hawsawi et al., (2017) [14].

To make EEG-fMRI recording safe, adjustments need to be made on both the EEG and MRI sides. In addition to meeting the general safety requirements for EEG equipment used for human participants, designing an EEG system that is intended to be used in the MR environment must take into consideration factors that are not considered in a regular lab or clinical environment. The static magnetic field used in MR requires that non-ferrous materials and components must be used. Furthermore, the electrodes and electrode lead wires are vulnerable to the heating effects described above. The potential risk for heating can be reduced by including current-limiting resistors in the electrodes [11], although it should be noted that this does not offer complete protection from heating [15]. EEG lead wires are usually made from copper, and the effects of the EEG on the electromagnetic field of the RF coil are affected by lead resistivity [15] so using materials that are less conductive than copper has also been investigated. For example, carbon fibre leads and a new conductive ink technology have been shown to heat less than copper during MRI scans [16]. This also has a potentially beneficial effect on the MRI images (see Section 3.2.2). The number of electrodes and the configuration of the electrode cap is also important, for example it has been found that the number of scalp electrodes influences the degree of heating [13] and less heating has been observed when cables are positioned along the *z*-axis of the scanner [17]. These findings have been integrated into the design of EEG systems and their associated guidelines for use, illustrating that technological develops can be driven by the requirements and findings of the research community.

From the MRI perspective, adapting the setup or imaging sequence is necessary to reduce the risk of adverse events and careful selection of the MR sequence used in combination with EEG is perhaps the most important safety factor to consider. MRI sequences utilising low RF power are recommended for EEG-fMRI because heating has been shown to increase linearly with RF deposition [18]; however, constraining the fMRI sequence can have implications for the MR data in terms of temporal or spatial resolution. This is an example of where compromise might be necessary to accommodate the EEG system in the scanner. Such a situation becomes more problematic as fMRI sequences advance and become more powerful and to take full advantage of the benefits offered by new, faster sequences, the EEG system must be able to function safely under those conditions. Often in multimodal imaging one modality progresses and the other must ‘catch up’ or develop in parallel. fMRI is a younger and potentially more dynamic modality than EEG, therefore, new developments are coming faster for fMRI technology than EEG technology. A recent example that has presented challenges for the EEG system is the development of multiband (MB) fMRI. The potential for increased RF power deposition during MB sequences means that safety becomes a critical issue if simultaneous EEG-fMRI is to remain at the cutting edge of neuroimaging research. Recent studies have investigated MB fMRI sequences for EEG-fMRI [19,20], showing that it can be performed safely within a set of specific sequence parameters. While this shows that current EEG systems can be used with advancing MRI techniques, it highlights the importance of safety testing and points to the need for the further development of EEG systems so they can continue to be used with advancing MR technologies.

Safety testing is critical to the development of EEG-fMRI. Temperature measurements in phantoms and humans help to assess the magnitude of heating effects and determine the conditions under which EEG-fMRI can be performed safely; however, it can be difficult to position the temperature probes in the places showing the most heating and to test all possible configurations. Electromagnetic (EM) simulations provide high-resolution estimates of the specific absorption rate (SAR) distribution across the head and this approach has been used to assess the likely heating associated with a specified EEG-fMRI setup [15,21,22]. This is particularly useful to model different configurations of the EEG system in the scanner and different sequence parameters. Developing appropriate models for EEG systems can speed up the research into different configurations and parameters so that empirical measurements can be more effective.

While the MR environment does present some safety concerns for concurrent EEG measurements it is worth noting that simultaneous EEG-fMRI has been carried out in many MRI centres worldwide with minimal adverse events. In their review paper, Hawsawi et al. [14] noted that no significant injury linked to EEG-fMRI has been reported to European or North American authorities and that EEG-fMRI is now considered a standard imaging technique. This is testament to the thorough investigation of the phenomena described above by the research community and the implementation of such findings by commercial manufacturers.

### 3.2. Data Quality

In addition to safety considerations, we must acknowledge that both the EEG and fMRI data are affected by the presence of the other measurement modality. The effects of the MR environment on the EEG data are greater than the effects of the EEG system on the MRI data, consequently, there has been more research on the former than the latter.

#### 3.2.1. EEG Data Quality in the MR Environment

Artifacts similar to those seen in a lab environment can also be expected in EEG recorded in the scanner, for example, eye movements, muscle tension, or electrical interference from devices used within the scanner room; however, there are artifacts that are specific to the MR environment that need special attention. These artifacts are predominantly due to electromagnetic induction, as described by Faraday’s law. Changes in the magnetic field over time or movement of the components of the EEG system inside the static magnetic field can induce a voltage in the EEG recording circuit. The main artifacts of concern are the gradient artifact, pulse related artifact, and motion related artifact. Each is described in turn below along with the methods commonly used for removing these artifacts. A more detailed overview can be found in a systematic review by Bullock et al. [23], including some recommendations for minimising and handling MR related artifacts in the EEG data.

Gradient artifact (GA). The application of time varying gradients during scanning induces electromotive forces in the EEG system, specifically in the loop formed by the electrodes and lead wires connected to the participant. These induced voltages from the gradient switching are referred to as the gradient artifact (GA). The GA has an amplitude hundreds of times larger than the EEG [24] and has multiple frequencies, some of which overlap with relevant EEG frequencies. For these reasons, removing the gradient artifact is essential to retrieving a usable signal from EEG recorded in the MR environment. 

The earliest EEG-fMRI studies avoided GA by using sparse, or interleaved, fMRI sequences whereby the acquisition time is shorter than the TR, allowing for a short period of time where the EEG is not affected by the GA. Indeed, for studies of gamma oscillations, sparse sampling sequences are still sometimes used because the frequency characteristics of the residual GA overlap with the signal of interest, particularly the high gamma range (>30 Hz) [20,25]. EEG triggered acquisitions have also been used to obtain EEG data with no GA [26]. Nevertheless, for a fully integrated experiment, appropriate conditions are needed for successful removal of the GA; this includes EEG systems capable of measuring these signals and correction strategies to remove the artifact from the recorded EEG signal.

Several EEG system parameters should be considered with respect to the GA. The amplitude of the induced artifact is proportional to the area of the loop formed by the EEG electrodes and lead wires. Consequently, the lead wires on EEG caps should be carefully routed and bundled together upon leaving the cap to reduce the loop size and help reduce the amplitude of the GA [27]. Reducing the length of the cables can also be beneficial [28], while some amplifier specifications also need to be considered. The RF used during scanning is very high frequency and the GA also has high frequency components; it is preferrable to filter these high frequency signals on the hardware level at the amplifier front end [29,30], therefore, a low pass filter should be implemented. A fast sampling rate is required to sample the rapidly changing components of the GA so that template-based correction strategies can be used for removing the GA [31]. Typically, a sampling rate of 5 kHz is used for recording EEG in the scanner which is much higher than that needed for EEG signals. The dynamic range of the amplifier must be sufficient to capture the GA without saturating the analogue to digital converter [31]. GA has an amplitude much larger than the EEG signal, therefore, the amplitude resolution must be determined accordingly. Commercially available EEG systems have been designed to specifically fulfil these criteria and in combination with the artifact correction strategies outlined below, meaningful EEG data can be successfully retrieved from data recorded during an fMRI sequence.

Average artifact subtraction (AAS) was introduced to the field of EEG-fMRI in 1998 [24] and it is still the most widely used approach for GA correction [23] (Figure 2a). AAS averages the GA over many imaging volumes and this average is then subtracted from the recorded data on a channel-by-channel basis to leave the cleaned EEG. To make sure that the average artifact template closely matches the artifact to be subtracted, precise timing and recording of the artifact is necessary. It has been shown that residual GA can be reduced by synchronising the EEG acquisition with the scanner clock [32] and this has been implemented as standard in the most commonly used commercially available MR compatible EEG systems. It has also been shown that matching the MRI sequence parameters to the sampling frequency of the EEG helps to reduce the residual artifact and it is recommended that the TR (repetition time) and slice duration should be an integer multiple of the duration of the EEG sampling time (i.e., 200 µs when sampling at 5 kHz) [33].

While AAS works satisfactorily under many circumstances [34], its effectiveness is reduced if the GA varies across time, from volume to volume. This is particularly important in studies where movement might be integral to the study, for example, those involving motor tasks or studies of patient populations where movement is more likely, e.g., epilepsy and Alzheimer’s. Using a slice rather than volume-based template subtraction method can help when the GA is affected by motion [35]. Some modifications of the AAS to reduce the residual artifact have been investigated, such as the inclusion of realignment parameters representing head motion [36] or weighting of the templates [37,38]. Other approaches to GA correction have also been considered, such as included blind source separation methods like an optimal basis set (OBS) [39] and independent vector analysis [40], while a beamformer spatial filter has been shown to supress the residual artifact [41]; however, many advanced artifact handling methods that are developed by research groups fail to be adopted by the wider community and AAS remains the most commonly used [23]. Optimally handling movement related effects on the GA remains an open problem.

Pulse-related artifact (PA). PA is also commonly referred to as the ballistocardiogram artifact or the cardioballistic artifact. PA is the result of an interaction between the static magnetic field and the cardiac cycle of the participant, and several mechanisms contribute to PA [42,43]: 1. artifact is created due to the flow of an electrically conductive fluid (blood) (Hall effect); 2. small movements of components of the EEG system (electrodes and lead wires) caused by the heart beat result in artifact due to induction; and 3. pulsing (movement) of the arteries causes artifact due to induction. PA is potentially more problematic than GA due to its non-periodic variability across time and that it is close to the frequency and amplitude of the EEG signals of interest. Furthermore, PA can vary in morphology across channels, the amplitude can be markedly different across individuals, and the effect scales with static field strength [44,45]. Variability in the PA can also be caused by variations in head position and fluctuations in the respiratory and cycle and heart rate [46]. Additionally, there is no temporal marker indicating the onset of each heartbeat, therefore, PA correction usually requires the time course of an ECG signal to determine the onset of each event. This can be problematic because the ECG signal itself is distorted in the magnetic field. Altogether, these factors result in an artifact that is variable across time and subsequently less easy to handle with a template subtraction algorithm.

However, despite PA variability, AAS has also been used to correct for PA artifact [23,24] and is still the most common method used for PA correction in the literature [23] (Figure 2b). Other methods for correction have been investigated, for example, a canonical correlation approach [47], harmonic regression [48], OBS [39], and ICA [49,50]. Studies comparing the efficacy of different methods have concluded that a combination of two methods can be helpful, for example, channel-wise correction using AAS or OBS followed by ICA to remove the residual artifact [50,51]. While several methods have been investigated, AAS is still the most common, and due to a lack of comprehensive comparison studies that evaluate the reduction of PA and preservation of the EEG signal, there is still no ‘one size fits all’ optimal solution for correcting a pulse-related artifact, although carbon wire loops offer a potential solution (see below).

Motion related artifact. Other artifacts observed in the MR environment are generally caused by movement of the electrodes and/or lead wires in the magnetic field. Some are the result of vibrations associated with a number of factors that can be present in the scanner room, for example, the helium pump [22,52,53] used for the supercooled magnet, the ventilation system in the scanner bore [54] or even lights or other electrical equipment in the scanner room [52]. These artifacts are best reduced by optimising the setup, e.g., using sandbags to stabilise cables and equipment and switching off any devices that can be safely switched off during scanning.

Potentially more problematic are the artifacts related to movement of the head, and therefore the electrodes and lead wires. We can do our best to stabilise the participant’s head during the MRI scan but there will be small movements throughout the experiment. This becomes even more relevant in special populations where movements are more likely or where the experiment involves a motor task. Given that head movements are not predictable or periodic, they cannot be handled with the same template subtraction approach used for the GA and PA. Eliminating the sections of data contaminated by motion artifact has been suggested but it is not an ideal solution due to the potential for substantial data loss. An alternative approach is to use additional hardware to record the motion related artifact independently of the EEG data, and then to use these channels as a reference signal for removing the artifact, for example using piezoelectric sensors [55] and more recently, loops to detect motion [56,57,58], a reference layer of electrodes [59,60], or Moire phase tracking markers and camera [35,61].

These methods are relatively new and have only recently gained some momentum in the literature. Motion is also problematic for fMRI data quality and prospective motion correction using cameras has been employed, therefore, Maziero et al. [35,61] used cameras intended for prospective motion correction to also correct motion related artifacts in EEG data recorded during fMRI. They were able to reduce the motion related voltages in the EEG thus making the EEG-fMRI more robust to motion and potentially opening up the technique to more difficult populations. While not yet widely adopted, this method holds some promise since it can address motion related issues in both the EEG and fMRI data. Carbon wire loops (CWLs) and reference layer subtraction have received a little more research attention so far. The principle behind these methods is that voltages will be induced in a loop or circuit moving in the magnetic field and measuring these voltages in a channel independent of, but close to, the EEG channels will allow the motion artifact to be measured independently of the EEG signal and then subsequently removed using regression. An additional advantage is that because the PA is also motion related, these methods can also be used to remove the PA and solve some of the problems described above when using template-based methods for PA correction. Indeed, it has been shown that reference signal methods perform better than template methods for removing both a PA and motion related artifact [62,63]. A study comparing motion correction methods showed that using a reference layer of electrodes was more effective than CWLs and Moire tracker methods, especially for smaller movements similar to those typically seen in EEG recorded in the scanner [62]; however, the reference layer is more difficult to implement practically than CWLs and as yet is not a commonly used method.

Given their advantage for handling several motion related artifacts, CWLs are now available as a commercial hardware solution with an analysis tool in a commercial software. This is an example of how innovation from the research community can result in standard solutions becoming available for the wider research community and highlights the need for cooperation between science and industry for multimodal imaging technologies to progress. It is anticipated that the use of CWLs will become standard practice for handling PA and motion related artifacts in EEG data recorded in the scanner.

#### 3.2.2. MRI Data Quality in the Presence of an EEG System

There has been less research into the effects of the EEG system on MR images, but some considerations are needed. First, RF emissions from the EEG system can interfere with the RF signals received from the investigated tissue if they fall within the detection frequency of the scanner; however, this effect can be reduced by using low power components and shielding the amplifier appropriately [30]. Perhaps more important is that both the static (B0) and oscillating (B1) magnetic fields required for MR imaging are affected by the presence of the EEG system. Given that image quality is dependent on both fields, it can be expected that image quality is affected by the presence of the EEG system in the scanner. The EEG system has different magnetic susceptibility properties than the human tissue being imaged and this difference in susceptibility causes perturbations on the B0 field. An extremely homogeneous B0 field is required for imaging, therefore, these B0 field inhomogeneities can cause image distortions and signal loss (dropout) [10]. The severity of these effects depends on the spatial orientation of the electrodes with respect to the B0 field, the field strength, and the difference in magnetic susceptibility [64]. The conductive materials of the EEG system can also affect the uniformity of the radiofrequencies (B1 field) used for scanning. Inhomogeneities in the strength of B1 around the object being scanned will result in intensity variations in the image. Additionally, exposure of electrically conductive components to the RF fields results in surface currents being generated, and these currents can also disturb the B1 field and cause signal loss [10]. This effect is worsened by B0 inhomogeneities [52], therefore the combined effects of the EEG system on the B0 and B1 fields must be considered. While it has been shown that the EEG system can have an effect on image quality, EEG-fMRI can be performed without compromising the image quality to the point where it is unusable because the effects are usually at the scalp level of the image and do not extend deeper [65,66]. It is worth keeping in mind, however, that the signal to noise ratio of an MR image can be reduced as the number of electrodes increases [67], so that higher density EEG recordings will potentially have more of an effect on the image quality.

### 3.3. Data Integration

After establishing a safe setup and optimising both the EEG and fMRI data quality, attention then turns to exploiting the information contained in both datasets. As discussed, the attraction of simultaneous EEG-fMRI is to take advantage of the complementary temporal and spatial resolution of each, but how exactly is that achieved? How can the data sets be integrated, or used together, to tell us more than we could gain from using a single measurement technique? The datasets can of course be analysed separately [68,69] but to take full advantage of multimodal datasets, analysis approaches for integrating the data are needed. There are several ways that EEG and fMRI data can be integrated depending on the nature of the question being answered, and here a general overview of the approaches will be given, for detailed reviews see [70,71,72].

EEG-fMRI data integration methods can be broadly divided into symmetrical approaches that integrate both modalities into a joint analysis and asymmetrical approaches where information from one modality is used to predict or constrain the other modality. The symmetrical approaches can be model driven or data driven, both of which have their advantages and disadvantages. On the one hand, model driven approaches take advantage of our understanding of complex neurovascular coupling dynamics to predict activation, but they require the explicit definition of the common neuronal process that elicits both EEG and fMRI signals [73,74]. Data-driven approaches on the other hand, avoid the need to model the complex neurovascular coupling dynamics, thus avoiding bias, and include methods such as joint ICA [75,76], using an information theoretic framework [77,78], or multivariate machine learning [79,80]. The asymmetrical approaches have been used more frequently in the literature and will be described in more detail below. To summarise, ‘EEG informed fMRI’ utilises the onset and duration of neuronal events measured in the EEG to identify brain regions where the fMRI signal is related to these events [81,82] (Figure 3), whereas ‘fMRI informed EEG’ is where areas of fMRI activation can be used to constrain the neuronal source estimation in the EEG [83,84] (Figure 4).

EEG-informed fMRI has received more research attention than fMRI informed EEG, presumably due to the rich temporal dynamics of the EEG data that can reflect different aspects of perception and cognition. There are several approaches to EEG-informed fMRI, with the methods being broadly divided into univariate and multivariate. In univariate models, a time course representative of the phenomenon of interest is obtained from a single (or limited number) of channels and then used to predict the BOLD signal. Multivariate techniques on the other hand, consider multiple EEG channels to capture spatial information, for example, functional connectivity measures across different channels or spatial correlation measures. EEG-informed fMRI has been used in several research fields, using a number of different EEG features (see Section 4 and Section 5 for an overview).

The goal of fMRI-informed EEG is to solve the spatial EEG inverse problem by guiding the source analysis using results obtained from the fMRI. The high spatial resolution of the fMRI BOLD response can be used to inform an EEG spatial model, allowing the researcher to characterise responses or changes on the timescale of the EEG. The spatiotemporal pattern gained from fMRI can be enhanced by the temporal resolution of the EEG derived information [85] and it has been shown that fMRI informed EEG can achieve better neuronal source reconstructions [83,86]. One of the reasons for there being less research into fMRI-informed EEG is that simultaneous measurements are not always necessary or recommended in the context of source reconstruction [70,87]. An interesting alternative to source localisation was demonstrated by De Martino et al. [79] who used multivariate machine learning-based regression to predict the power of EEG oscillations from simultaneously acquired fMRI data during an eyes-open/eyes-closed task. This study went beyond correlation of the two signals and showed that EEG-fMRI can be used to predict the signal in one modality from information from the other modality.

## 4. Applications of EEG-fMRI

### 4.1. What Can We Measure Using EEG-fMRI?

The EEG signal is traditionally analysed in terms of the power of rhythms in the spontaneous EEG or stimulus or task-specific event-related potentials (ERPs) [2]. The most relevant brain oscillations are found in the following frequency bands: delta (0.5–4 Hz), theta (4–8 Hz), alpha (8–13 Hz), beta (13–30 Hz), and gamma (above 30 Hz) [88]. ERPs are usually described in terms of the amplitude and latency of the peaks and troughs in the waveform and represent time-locked and/or phase-locked activation of neuronal populations in response to external stimuli, for example, sensory stimulation or cognitive tasks [89]. EEG is a more mature technique than fMRI, has a rich history of research and many standard measures representing different aspects of perceptions and cognition. Consequently, there is a multitude of possible EEG features to extract.

The BOLD fMRI signal is characterised by the haemodynamic response function (HRF), that accounts for the delay between a stimulus or change in state and the associated vascular response. Typically, in studies of neuronal responses a period of activation during a task is compared with a baseline, or rest period. This results in a spatial map of the brain regions that are active during the task. Advances in analysis techniques means that it is now also possible to measure intrinsic or spontaneous brain activity using BOLD fMRI. These advances were made to reflect the view that the brain is a network of functionally connected regions and that patterns of connectivity should be investigated as well as the localised, task related activity [90].

It is possible to measure event related and spontaneous activity with both EEG and fMRI. The myriad of possible measures in each individual measurement modality along with the many ways of integrating the data sets has resulted in the contribution of EEG-fMRI to many application fields. There are a several review articles that cover EEG-fMRI (e.g., [70,71,72,91]) so here I will only briefly cover the areas where EEG-fMRI has contributed to the literature and then focus on the emerging fields and future direction in the next section.

### 4.2. Event-Related Responses

The first study reporting a visual evoked potential in simultaneous EEG-fMRI was carried out in 1999 by Bonmassar et al. [92], and since this time a vast amount of EEG-informed fMRI studies have been performed using ERPs in sensory and cognitive modalities. An example of ERPs and an event related BOLD response for the same visual oddball paradigm can be seen in Figure 5. Considering all ERP applications is beyond the scope of this review but a particularly interesting approach in task-based studies is a single trial analysis of the ERP components. This allows access to the rapidly changing neural dynamics associated with perception, cognition, and performance to be modelled in the fMRI analysis, thus providing a spatiotemporal resolution that cannot be achieved with one method alone [82,93,94,95] (Figure 6). By using well established paradigms and designing experiments where manipulations of the paradigm can induce the modulation of behavioural and brain responses, Eichele et al. [94] were able to identify several spatially separated, event-related regional activations related to the processing stages of perceptual inference and pattern learning. Debener et al. [82] found the single-trial error-related negativity of the EEG to be systematically related to behaviour in the subsequent trial, reflecting adjustments of a cognitive performance monitoring system. This trial-by-trial EEG measure of performance monitoring predicted the fMRI activity in a brain region that plays a key role in processing, showing how the trial-by-trial EEG-fMRI response reflects behavioural changes. These studies showed how regressors representing the trial-to-trial variability of the EEG signal can provide more fine-grained temporal information that can be used to differentiate the roles of different cortical regions in a given task. This opened the door for many other investigations on cognitive function in healthy brains and disease states and can be considered a significant development. While these studies illustrate that the ERP informed fMRI analysis can be valuable, the paradigm eliciting the ERPs must be considered, specifically whether it can be used optimally for event related responses in both modalities [96]. It is also important to understand what the ERP component of interest represents when deciding what ERP component to include in the fMRI analysis [97].

### 4.3. EEG Frequency Content

EEG-fMRI has also made it possible to better understand the functional role of different EEG oscillations in both task-related and spontaneous measurements. Measuring spontaneous neuronal events and their haemodynamic correlates is something that can only be achieved by combining EEG and fMRI because separate sessions simply do not allow for the correlation of the two measures. This is particularly useful for investigating the relationship between EEG oscillations and the BOLD response, for example, in resting state networks and functional connectivity. An example of EEG alpha activity and corresponding BOLD activation can be found in Figure 7. The initial research focussed on the relationship between the alpha band power and haemodynamic changes in the brain. Discrepant findings on the relationship between the alpha power and the BOLD response [81,98] led to speculation that the wider frequencies should be considered to fully understand the brain at rest [99]. Indeed, subsequent work has shown that different frequency oscillations in the EEG are associated with different relations to BOLD changes at rest and during tasks. Mayhew et al. [100] found a significant interaction between the amplitude of spontaneous alpha-power and the magnitude of both positive and negative BOLD responses to visual stimulation. They suggest that there is a ‘baseline’ of spontaneous activity that can modulate the amplitude and shape of both a visual BOLD response and changes to resting state networks, hinting at a functionally connected network during the resting state. Scheeringa et al. [101] found that the neural processes underlying high-gamma power and those underlying alpha and beta power independently contribute to explaining a BOLD variance. Using an attentional monitoring task, trial-by-trial BOLD fluctuations were shown to correlate positively with trial-by-trial fluctuations in narrow-band high frequency (60–80 Hz) EEG gamma power but BOLD fluctuations correlated negatively with EEG alpha and beta power. This study also extends animal work on the coupling between BOLD and high-gamma band oscillations showing that a similar relationship holds in humans performing a cognitive task. It is not easy to replicate the intracranial studies performed in animals in human participants and being able to use non-invasive EEG-fMRI to probe this is a remarkable step forward in our understanding of the electrophysiological underpinnings of cognition-related haemodynamic responses in humans. Additionally, by using high resolution fMRI, Scheeringa et al. [25] demonstrated that changes in specific frequency bands in the EEG can be related to changes in the BOLD signal measured at different cortical depths. Variation of alpha band power across trials of a visual attention paradigm was related to the BOLD signal in both the deep and superficial layers while gamma band activity showed a relation to the BOLD signal only at the middle and superficial layers. This finding shows that different underlying neural dynamics contribute to the generation of the BOLD signal at different cortical depths. The authors suggest that combing high resolution laminar fMRI measures with EEG provide a potential method for linking human EEG-fMRI to systems-level neuroscience in animals, again supporting the use of advanced EEG-fMRI methods for characterising the network structure and the functional neurophysiological mechanisms involved in cognitive processing.

### 4.4. Epilepsy

One application where EEG-fMRI has received a lot of attention is the study of epilepsy. Indeed, the origins of simultaneous EEG-fMRI are in epilepsy research and this field has contributed to the development of the technique and analysis strategies [102,103,104,105]. Localisation of epileptic generators is a critical topic for the understanding and treatment of epilepsy; however, localising the precise region of epileptic foci is challenging, but simultaneous EEG-fMRI can be used to meet this challenge [106]. Electrical brain activity associated with epilepsy can be measured as interictal epileptiform discharges (IEDs/spikes) or seizures using scalp EEG [107]. These discharges cause an increase in metabolism and blood flow that can be measured with the BOLD signal. Using the time course of these discharges from the EEG can inform the fMRI analysis to reveal which brain regions generate the IEDs and sometimes identify more widespread responses associated with the IED, for example, how they propagate [103,104,105,108]. For a review, please see Shamshiri [109]. Figure 8 provides an example of IEDs in EEG data and BOLD responses to these IEDs. There have been fewer EEG-fMRI studies of seizures (ictal events) than interictal events due to seizures happening infrequently and unpredictably, and practical concerns about patient safety and motion related artifacts in the data [110]; however, EEG-fMRI does hold promise for a more thorough understanding of seizures [111].

In addition to furthering our understanding of the epileptic brain, simultaneous EEG-fMRI has clinical relevance in the pre-surgical evaluation of patients with drug-resistant focal epilepsy. The exact source of epileptic seizures cannot be visualised with fMRI alone, therefore invasive stereo-EEG analysis is often required. Simultaneous EEG-fMRI can offer a non-invasive alternative for the localising brain regions generating interictal epileptiform activity; however, if the technique is to become a routine part of presurgical evaluation, it must be able to locate the regions responsible for the epileptic events accurately and reliably [112]. This means comparison with the current gold standard of intracranial EEG is desirable, for example, comparing generators identified with EEG-fMRI with EEG data from the same patients later implanted with subdural grids or strips [113]. Alternatively, intracranial EEG electrodes can be used during fMRI, and while this is risky due to the potential for heating described in Section 3.1, work has been conducted using intracranial electrodes during fMRI [17,114]. Markoula et al. [115] investigated the impact of EEG-fMRI on the epilepsy surgery decision-making process in patients with refractory extratemporal epilepsy and found that EEG-fMRI had a significant impact on epilepsy surgery planning, going on to suggest the complementary information yielded by EEG-fMRI should be considered in presurgical evaluation.

Deficits in many cognitive domains are recognised as co-morbidities of epileptic disorders, supporting the idea that epilepsy is a network disease associated with complex cognitive deficits [116]. We have already discussed how EEG-fMRI can facilitate the study of resting state and cognitive tasks, it is therefore logical to look at the effects of epilepsy on cognitive function using EEG-fMRI. For example, decreases in resting state network activity have been found in temporal lobe epilepsy [117] and generalised epilepsy [103,118]. Laufs et al. [117] found that the brain areas that are active in a state of relaxed wakefulness are deactivated during IEDs of temporal lobe origin but not those of extra-temporal lobe origin. Gotman et al. [103] investigated generalised bursts of epileptic discharges and found both thalamocortical activation and suspension of the resting state networks. They suggest that these effects may combine to cause the state of reduced responsiveness observed in patients during spike-and-wave discharges. Shamshiri et al. [119] found that attentional network activity was reduced in epilepsy patients compared to controls and that task related activity was also reduced. These studies show that EEG-fMRI can be used to explore cognitive deficits in epilepsy patients (see Shamshiri et al. for a review [109]).

While simultaneous EEG-fMRI holds great promise for epilepsy research, many studies fail to reach comprehensive conclusions, some due to the absence of interictal epileptiform discharges during simultaneous recordings, and others because there is a lack of haemodynamic response correlated to the interictal epileptiform discharges [120]. This has led to methods being developed or considered to improve the measurement of epileptic activity. For example, flexible HRF models may be more appropriate for detecting responses due to possible differences in the delay or shape of the response in comparison to the canonical HRF [112]. Data driven rather than model driven approaches have also been considered to address this issue; using ICA of the BOLD signal to investigate HRF variability with respect to IED amplitude fluctuations was shown to improve the specificity of EEG-fMRI analyses [104]. Alternative methods have also been suggested for measuring the EEG signal of interest that is included in the EEG-informed fMRI analysis. For example, Grouiller et al. [120] constructed topographic maps from IEDs recorded during long-term video-EEG monitoring. These topographic maps were then correlated with the EEG recorded during the fMRI session and used to evaluate BOLD changes associated with epileptic activity. Their findings showed a correlation between the epilepsy-specific EEG maps and BOLD changes. They were also able to validate their findings against intracranial EEG for localisation of focal epileptic activity. Ebrahimzadeh et al. [121,122] proposed a novel method whereby information contained within the entire time series of a relevant EEG source was used in the analysis of the EEG-fMRI data by using independent component analysis (ICA). This approach addresses a potentially important limitation of the conventional approach where only information at the time of the IEDs is considered, ignoring neural activities that could be associated with epileptic generators at other time points. The authors suggest this method creates a more realistic perception of the neural behaviour of epileptic generators and is superior to conventional EEG-fMRI approaches. It will be interesting to see how this method develops.

In summary, EEG-fMRI has contributed much to the understanding of epilepsy and has the potential to be used in routine clinical use. Additionally, EEG-fMRI has benefited greatly from the epilepsy research community pushing forward with this technique.

### 4.5. Sleep

Sleep is fundamental to the healthy function of the human brain; however, the mechanisms and functions of sleep are not yet fully understood. Nevertheless, EEG has contributed enormously to the understanding of sleep and there are clear EEG signals associated with different sleep stages. In fact, there are international standards for sleep scoring based on EEG, EMG, and EOG [123]. Other electrophysiological measures such as EMG and EOG, and measures such as pulse oximetry and airflow, are also often measured together with EEG during the assessment of sleep states. This multimodal approach is known as polysomnography (PSG) [124]. Measuring peripheral physiology signals in the scanner can be challenging, due to the safety and data quality issues described earlier. Often EMG or EOG measurements require longer lead wires than EEG that can be susceptible to heating and scanner related artifact; however, it is possible to measure peripheral physiology alongside EEG in the scanner making polysomnography measurements achievable [125,126].

Using fMRI alone to study sleep can be difficult because it is not easy to determine whether a participant is indeed asleep in the scanner, only perhaps from an absence of behavioural reactions. Therefore, in many fMRI studies there is some ambiguity over whether the measurements are of sleep states or not. Given that EEG can indicate when a participant is asleep, and can show different sleep states, the combination of EEG (or other PSG measures) with fMRI makes sense and has allowed further investigation of the neural networks underlying sleep. In a similar way to that described above for epilepsy, EEG provides a way to access specific temporal events or changes in neural activity that can then be used in the fMRI data analysis.

EEG-fMRI has been used to show that there is a specific pattern of synchronised decreased brain activity during sleep [127]; that a specific network is associated with regulation of the sleep vigilance level and arousability that reflects sleep instability [128]; and that specific brain regions are active during sleep spindles in non-rapid eye movement sleep [125,129]. Additionally, the thalamo–cortical network has been identified as being important in the transition from sleep to wakefulness and that functional connectivity within the thalamic regions also changes across the sleep–wake cycle [130]. Interestingly, it has been shown that this thalamo–cortical network is disrupted in some disorders such as idiosyncratic generalised epilepsy [131], and insomnia disorder [132]. An example of sleep EEG and the BOLD correlates of sleep spindles can be seen in Figure 9.

Sleep has also been associated with effects on various aspects of cognition and memory, and EEG-fMRI affords the opportunity to investigate this further. It has been shown that a subset of brain activations time-locked to sleep spindles are specifically related to reasoning abilities but were unrelated to short-term memory or verbal abilities [133]. Sleep spindles have also been associated with memory consolidation, and Jegou et al. [134] found evidence for a cortical reactivation during fast spindles, especially in regions involved in learning before sleep. These regions were also activated during recall after sleep, suggesting a role for these patterns of cortical activation in memory consolidation.

Some interesting recent research has also included the measurement of cerebrospinal fluid (CSF) in addition to EEG-fMRI. By combining EEG with a fast fMRI sequence that can detect fluid inflow as well as blood flow, the CSF can also be measured. CSF clears metabolic waste products from the brain and it is not known whether this process is related to brain activity during sleep. Fulz et al. [135] found that CSF dynamics are interlinked with neural and haemodynamic rhythms, specifically that slow neural activity is followed by brain-wide pulsations in blood volume and CSF flow. This suggests that EEG slow waves may also be linked to the restorative effects of sleep.

It is worth noting that EEG-fMRI sleep recordings can be challenging in the MR environment due to the noise of the scanner and comfort of the participant. Additionally, extended recording times are often needed which exacerbate the comfort issue, especially when the participant is also wearing an EEG cap; however, the research described above illustrates that it is possible to perform high quality sleep research with EEG-fMRI and that this multimodal approach has enabled the progression of our understanding of sleep.

## 5. Emerging Fields for EEG-fMRI

### 5.1. Neurofeedback

EEG-fMRI can also be used to regulate brain activity rather than just to measure it, for example, using neurofeedback. Both EEG and fMRI have been used separately in neurofeedback [136,137] but their combination in this context is relatively new. Neurofeedback uses neuroimaging techniques, such as EEG and fMRI, to acquire real-time measures of brain activity that can be used to encourage self-regulation of the targeted brain activity, with the goal of promoting behaviour modification. Neurofeedback opens new therapeutic possibilities in the fields of psychiatry and neurology by encouraging patients to learn self-regulation of the disordered brain regions. Simultaneous real-time fMRI and EEG neurofeedback combines real-time feedback from both modalities to simultaneously regulate both haemodynamic and electrophysiological brain activities [138,139].

Given what we have learned in Section 3.2 about the effects of the MR environment on EEG data, it is no surprise that real-time removal of those artifacts presents a challenge. The signals need to be processed quickly so that meaningful feedback can be given to the participant. Integration of such a strategy for real time EEG and fMRI is described in a proof-of-concept paper by Zotev et al. [139]. By making use of the available fMRI and EEG software and integrating it with their novel real time system, they were able to demonstrate the feasibility of simultaneous self-regulation of both haemodynamic and electrophysiological activities. Their multimodal neurofeedback graphical user interface can convert the cleaned real time EEG and fMRI data streams into graphical representations that can be viewed by the participant. This integration of the two data streams to make feedback available to the participant in a meaningful way is vital for the success of EEG-fMRI neurofeedback. Mano et al. [138] describe in detail a general method for setting up a hybrid EEG and fMRI platform for neurofeedback experiments and provide some guidance on the minimal technical requirements or features to look for when setting up EEG-fMRI neurofeedback. The group have subsequently used their EEG-fMRI neurofeedback system to successfully investigate motor imagery [140,141].

Multimodal neurofeedback offers some potential benefits for the treatment of neurological and psychiatric disorders. Targeting disorder-specific brain activity identified by two very different imaging modalities might have stronger effects than standalone applications of either method. In addition, real time EEG-fMRI neurofeedback training may help to develop personalized mental strategies by engaging both the fMRI and EEG target brain activities. These strategies can then later be used in more cost-effective EEG only neurofeedback or in therapeutic strategies. So far EEG-fMRI neurofeedback has been used in major depressive disorder (MDD) [142], and stroke rehabilitation [140], and EEG has been measured simultaneously with fMRI neurofeedback in post-traumatic stress disorder (PTSD) [143]. Zotev et al. [142] investigated emotion self-regulation in patients with MDD using real time EEG-fMRI neurofeedback. The participants simultaneously upregulated two EEG and two fMRI neurofeedback target measures related to major depressive disorder while inducing happy emotions. During the EEG-fMRI neurofeedback procedure, the MDD patients were able to significantly increase the frontal alpha and beta asymmetry and increase functional connectivity in target fMRI measures. Notably, the MDD participants showed significant mood improvements, including reductions in state depression, anxiety, confusion, and total mood disturbance, and an increase in state happiness after the neurofeedback training [142], suggesting that real time EEG-fMRI neurofeedback may have potential for treating MDD. Lioi et al. [140] applied EEG-fMRI neurofeedback using motor imagery for upper limb recovery in stroke-patients. Their results showed the feasibility of EEG-fMRI neurofeedback in motor recovery and the patients were successful in upregulating the activity of target motor areas; however, the effects were dependent on the stroke characteristics and motor impairment severity. Meanwhile, Zotev et al. [143] used fMRI neurofeedback to upregulate amygdala activity during a happy emotion inducing task in PTSD patients. The experimental group showed a significant reduction in PTSD severity and comorbid depression severity after the neurofeedback training. The EEG was recorded concurrently with the fMRI neurofeedback allowing the authors to investigate electrophysiological correlates of the fMRI-based neurofeedback training. This shows that simultaneous EEG-fMRI can be useful in neurofeedback studies even when the feedback itself is based on only one of the modalities.

Together, these studies in different patient populations suggest a role for EEG-fMRI neurofeedback in the treatment of neurological and neuropsychiatric disorders. Additionally, EEG-fMRI neurofeedback is a nice example of the progression from proof of principle to the potential clinical application of a new technique in a relatively short space of time. Table 1 provides an overview of potential clinical applications for EEG-fMRI when combined with other techniques.

### 5.2. Ultra-High Field EEG-fMRI

Structural and functional brain imaging at ultra-high fields (UHF) has become an established imaging method in recent years as the number of 7 T scanners installed worldwide increased [144]. Higher static field strengths offer better signal-to-noise ratio and increased spatial resolution of the fMRI data compared to lower field strengths. Adding the temporal resolution of EEG is an exciting prospect to fully exploit multimodal imaging at UHFs; however, at higher static field strengths, the safety risks and effects on data quality need to be reconsidered, it is not as simple as moving an established 3 T EEG-fMRI setup to a 7 T scanner. UHF MRI scanners require a higher radiofrequency system and these higher RF frequencies influence how the EEG system interacts with the RF transmit coil. As described in Section 3.1, this can have consequences for both safety and image quality [11,13]. RF power deposition increases quadratically with respect to magnetic field strength or resonant frequency, therefore the risk of heating can be higher at UHFs. Antenna effects need to be considered with respect to the different RF wavelength used at UHFs compared to 3 T and this influences how electrode lead wire lengths interact with the RF fields. Accordingly, the practical setup needs to be considered, specifically with respect to the availability of head coils suitable for EEG-fMRI at 7 T. Making a head coil that functions well for 7 T imaging does not necessarily easily accommodate an EEG cap and cables, presenting technical challenges for the manufacturers of head coils and EEG caps. Eddy currents can also be more problematic at higher field strengths because RF power has an inhomogeneous distribution that can result in unpredictable hotspots. Static field effects on the data are also a concern. For example, a movement related artifact (including pulse related artifact) is likely to be more problematic in EEG data and a susceptibility artifact from electrodes and cables could be concerning for MR image quality.

Consequently, EEG-fMRI research at UHFs has so far focused largely on addressing these difficulties. The safety aspects have been assessed [15,22], artifacts in both EEG and fMRI data have been investigated [16,22,44,46,64,145,146], and analysis methods for retrieving the signals of interest have been considered [147,148,149]. EEG and fMRI recorded at 7 T can be seen in Figure 10. Only recently has 7 T EEG-fMRI been used to explore questions about brain function in a small number of studies on sleep [150], epilepsy [151], and functional connectomes [152]. Now that the technical, safety, and data quality issues have largely been addressed and 7 T scanners are becoming widely available, it is anticipated that the interest in UHF EEG-fMRI will continue to grow. It is worth noting that most UHF EEG-fMRI studies have focused on 7 T except for a small number of institutions where a 9.4 T scanner is available [45,153]. The existence of a human 9.4 T scanner suggests that multimodal imaging could progress to field strengths higher than 7 T but it remains to be seen whether 9.4 T imaging in humans will become widely adopted. In any case, the challenge will be to develop EEG systems that are capable of keeping up with further developments in ultra-high field imaging.

### 5.3. Low Field EEG-MRI

While UHF brain imaging has progressed recently and combining it with EEG is a hot topic for current and future developments, it is also worth considering the role of low field MRI in multimodal brain imaging. Recent developments in portable low field MRI have shown that brain imaging is possible at the patient’s bedside in intensive care settings [154,155]. Low field MRI could enable the assessment of neurological injury in other scenarios such as the emergency department, mobile stroke units, and resource-limited environments [154]. Given that EEG is also a low-cost method and is often used in clinical diagnosis, the prospect of combining low field MRI and EEG is an interesting one. The safety and data quality issues that affect the EEG system at higher fields would not be problematic at very low fields, therefore, the technical issues are likely to be less challenging; however, the complementary nature of the signals is not as clear given that the aim of low field MRI is primarily structural imaging for diagnosis, whereas functional imaging is more likely to provoke the interest of the research community. Nevertheless, the benefits of concurrent low field MRI and EEG in a clinical setting could be worthy of investigation. So far, the author is unaware of published work using EEG and low field MRI and the specific benefits of the combination are not yet clear, but the prospect is an interesting one.

### 5.4. EEG-fMRI and Concurrent Brain Stimulation

Stimulation methods that allow the manipulation of brain activity as an independent variable make it possible to go beyond correlational analyses and investigate causal dependencies. Using simultaneous EEG-fMRI in combination with brain stimulation methods is an exciting prospect and is one that is gaining momentum.

TMS (transcranial magnetic stimulation) is a method for stimulating the brain using rapidly changing magnetic fields. It can be used to investigate whether a brain area is causally relevant for a cognitive task and to map the brain’s functional connectivity profile. TMS has therefore been widely used in cognitive neuroscience research and has been used in clinical practice for treating neuropsychiatric disorders such as depression and schizophrenia (see [156] for a review). EEG-fMRI is a powerful tool for investigating brain networks and combining it with TMS could provide additional information on causal dependencies within these networks [157]. TMS allows the controlled stimulation of one network node, EEG has the temporal resolution to capture fast neuronal fluctuations, and fMRI allows TMS-evoked propagation patterns to be monitored with high spatial resolution [158].

As with all multimodal imaging methods there are technical challenges to meet. Regarding the EEG data quality, artifacts from the stimulator need to be considered along with any potential interaction with other artifacts in the MRI environment. Peters et al. [157] demonstrated that TMS-EEG-fMRI measurements are feasible in terms of hardware and MR signal quality; however, the EEG data quality is affected by the TMS pulse and useful EEG signals were limited to pre-TMS intervals and EEG components arising 300 ms or later than the TMS delivery. Despite the effects of the TMS artifact, the EEG data can still be highly informative. Peters et al. [158] were able to map whole-brain TMS signal propagation using fMRI as a function of the pre-TMS oscillatory state measured with simultaneous EEG. This illustrates how the combination of brain imaging and neuromodulation techniques can reveal causal dependencies between oscillatory mechanisms and functional connectivity. This opens exciting possibilities for research into the dynamics of cognitive circuits.

Transcranial alternating current stimulation (tACS) applies frequency-specific sinusoidal electric currents through the scalp, and these currents can mimic and entrain endogenous oscillations. Combining such a stimulation method with EEG-fMRI can allow investigation of the temporal and spatial dynamics of brain networks. Clancy et al. [159] manipulated alpha oscillations using alpha frequency tACS targeting the occipitoparietal alpha source and measured the concurrent changes in alpha synchrony (EEG alpha power and connectivity) and default mode network connectivity (BOLD fMRI) before and after tACS using simultaneous EEG-fMRI. They showed that tACS of the alpha source in the occipitoparietal cortex augmented the alpha oscillations and strengthened the BOLD and alpha-frequency oscillatory connectivity. This supports the idea that the intrinsic connectivity networks have an electrophysiological origin underpinning them and clearly shows the benefits of neuronal and haemodynamic measurements in response to the same stimulation.

The combination of EEG-fMRI with brain stimulation techniques is relatively new but the progress made so far is promising and the possibility to further probe brain networks is likely to drive the application forward in the coming years.

### 5.5. Trimodal Imaging: EEG-fMRI in Combination with PET

Now that the major challenges of EEG-fMRI have been solved, the inclusion of a third measurement modality seems appealing. For example, the combination of EEG-fMRI with PET has received recent attention, especially with the introduction of hybrid PET-MR scanners to the market. The combination of MRI and PET technology in one scanner makes it easier to implement trimodal imaging and is a good example of where developments in one technique (PET-MR) encourages, or enables, the progression to other multimodal combinations. The advantage of combining PET with EEG-fMRI is being able to also consider metabolic aspects and measure multiple factors contributing to neuronal activity such as glucose consumption and the balance of inhibition/excitation and oscillations of brain activity [160,161] (Figure 11); however, adding a third measurement modality to the mix can potentially cause interactions that will affect data quality. Measuring all three modalities in one session has been demonstrated [162]; however, while highlighting the advantages of combining the three modalities, the measurements were not simultaneous but rather completed pairwise to minimise the effects of one modality on the others, for example, EEG electrodes on the PET images. Pioneering proof of principle work on truly simultaneous EEG-fMRI-PET measurements has shown that the effects of the EEG equipment on the attenuation of PET signals are negligible [163]. Furthermore, the EEG data is not negatively affected by the PET system, the dominant artifacts are those related to the MRI scanner [45] and as described above these can be satisfactorily corrected.

With the technical challenges under control, researchers have begun to utilise this trimodal approach to investigate aspects of brain function. Golkowski et al. [164] used EEG-fMRI-PET to investigate brain activity in patients with disorders of consciousness. They argue that combining these methods will provide multiple markers of consciousness under the same measurement conditions, thus enabling a more accurate clinical diagnosis and better estimation of prognosis in a condition that can be difficult to accurately categorise. Although their sample size was small, their finding that these methods can indeed be used together to give different measures of consciousness does suggest that trimodal imaging is advantageous in this context; however, more work is needed to be able to use a trimodal approach routinely to improve the diagnosis and prognosis of diseases of consciousness. The trimodal approach will also be valuable for investigating the fundamentals of brain function, for example, resting state networks that underpin many processes can be altered in disease states. By measuring the EEG-fMRI-PET of resting state networks, Shah et al. [161] were able to demonstrate the value of such an approach in human studies, and argue that “the trimodal approach holds great promise due to the richness and complexity of the data acquired in different dimensions and time scales; this has enormous potential for the development of individual biomarkers in form of the multimodal fingerprint for individual prognosis and individualized treatment planning e.g., in schizophrenia, depression and in the mid-term, potentially, dementia”. Recent work [165] has shown that event related potential components that are affected by disease states and disorders can also be useful in this trimodal approach. Task-induced changes in glutamatergic neurotransmission using the mis-matched negativity (MMN) paradigm were found, providing further evidence that EEG-fMRI-PET is a useful approach for investigating the role of glutamatergic neurotransmission in healthy participants and in patients with various disorders.

**Table 1 sensors-22-02262-t001:** Overview of potential clinical applications for EEG-fMRI combined with other methods.

Additional Method/Combination	Reference	Clinical Application	Main Finding
Study population/focus of study is a clinical group or specific disorder			
EEG-fMRI + Neurofeedback	[143]	Post traumatic stress disorder (PTSD)	The experimental group showed a significant reduction in PTSD severity and comorbid depression severity after the neurofeedback training.
	[142]	Major depressive disorder (MDD)	During EEG-fMRI neurofeedback, MDD patients were able to significantly increase frontal alpha and beta asymmetry and increase functional connectivity in target fMRI measures. The MDD participants showed significant mood improvements, including reductions in state depression, anxiety, confusion, and total mood disturbance, and increase in state happiness after the neurofeedback training.
	[140]	Stroke	Patients successfully upregulated the activity of target motor areas; however, the effects were dependent on the stroke characteristics and motor impairment severity.
EEG-fMRI-PET	[164]	Disorders of consciousness	Combining these methods provided multiple markers of consciousness under the same measurement conditions, thus enabling a more accurate clinical diagnosis and better estimation of prognosis in a condition that can be difficult to accurately categorise.
	[165]	Diseases that affect glutamatergic neurotransmission (e.g., schizophrenia and dementia)	Task-induced changes in glutamatergic neuro-transmission using the mis-matched negativity (MMN) paradigm were found showing that EEG-fMRI-PET is a useful approach for investigating the role of glutamatergic neurotransmission in healthy participants and in patients with disorders such as schizophrenia and dementia.
Potential clinical applications			
EEG-fMRI TMS	[158]	Cognitive deficiencies	The combination of brain imaging and neuromodulation techniques can reveal causal dependencies between oscillatory mechanisms and functional connectivity. This opens exciting possibilities for research into the dynamics of cognitive circuits.
EEG-fMRI tACS	[159]	Disorders where disrupted connectivity	The tACS of the alpha source in the occipitoparietal cortex augmented the alpha oscillations and strengthened BOLD and alpha-frequency oscillatory connectivity. This supports the idea that the intrinsic connectivity networks have an electrophysiological origin underpinning them and clearly shows the benefits of neuronal and haemodynamic measurements in response to the same stimulation.
EEG-fMRI + additional MR measures (CSF)	[135]	Sleep disorders	CSF dynamics are interlinked with neural and haemodynamic rhythms, specifically that slow neural activity is followed by brain-wide pulsations in blood volume and CSF flow. This suggests that EEG slow waves may also be linked to the restorative effects of sleep.

The work in this field so far is encouraging and the EEG-fMRI-PET trimodal approach holds potential for furthering our understanding of brain function.

## 6. Conclusions

Simultaneous EEG-fMRI is now a well-established neuroimaging technique. Considerable hardware developments and safety investigations have enabled the concurrent measurements of EEG and fMRI. Much work has been undertaken on investigating the nature and source of artifacts in both modalities and artifact handling methods have been implemented to make retrieving meaningful data from both modalities possible. Data integration strategies have been developed to exploit this rich source of information on the neural dynamics of brain function and these developments have led to a better understanding of cognitive processes and the ability to investigate brain function in ways that are not possible when using one measurement modality in isolation. Furthermore, the concurrent measurement of EEG-fMRI has demonstrated the potential of the technique to facilitate the development of clinical interventions. This review has shown that EEG-fMRI is still very much an active field of research with both well-established applications and applications still in their infancy that hold great promise for the future. Simultaneous EEG-fMRI is one of the most powerful tools available for the non-invasive study of human brain function and it is anticipated that the technology will continue to develop to support advanced research applications.

## Figures and Tables

**Figure 1 sensors-22-02262-f001:**
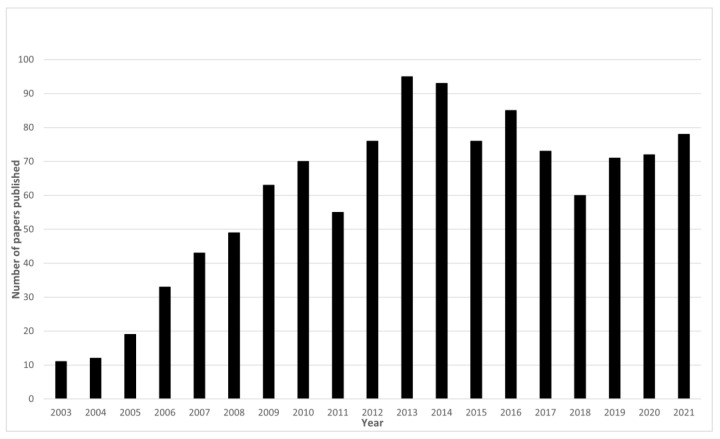
Number of EEG-fMRI publications per year. Numbers obtained from PubMed using the search terms EEG-fMRI (AND date range) and EEG/fMRI (AND date range).

**Figure 2 sensors-22-02262-f002:**
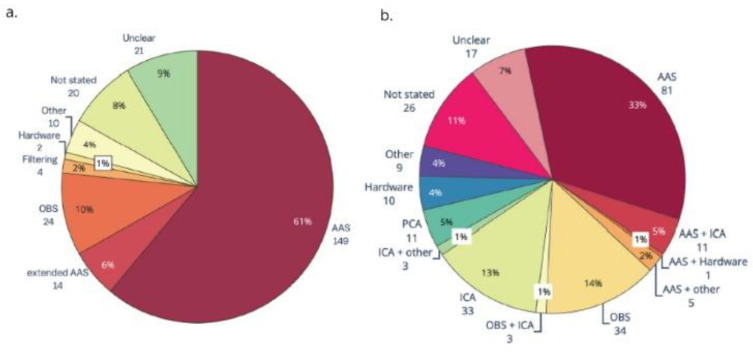
Part (**a**). Gradient artifact removal in EEG-fMRI papers, published between 2016 and 2019 (*n* = 244). AAS, average artifact subtraction; OBS, optimal basis set. Part (**b**). Ballistocardiogram (BCG) artifact removal method in literature using EEG-fMRI published between 2016 and 2019 (*n* = 244). AAS, average artifact subtraction; OBS, optimal basis set; ICA, independent components analysis; PCA, principal components analysis. Reproduced under open access license agreement CC BY 4.0 from Ref. [23]. Copyright 2021 the authors.

**Figure 3 sensors-22-02262-f003:**
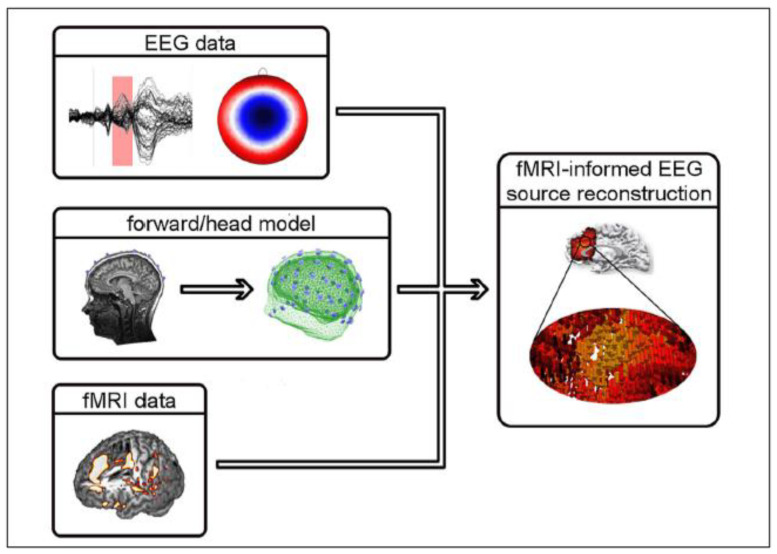
Illustration of fMRI-informed EEG source reconstruction. To estimate the location and activity of active cortical patches in the brain that lead to measurable EEG signal changes on the scalp, forward or head models are constructed from individual MR images. Here, volumes representing skin, skull, and brain tissue have been extracted. Based on such a model and the EEG time courses as well as corresponding scalp topographies (the pattern of EEG activity as recorded on a participant’s head), the EEG sources can be inferred (inverse modelling). Statistical maps from a standard fMRI analysis are used to further constrain the possible source constellations. The procedure used here for inverse modelling computes a high number of dipolar sources distributed across the brain, each of which is characterized by its position, orientation (pointing direction of an arrow), and strength (as indicated by colouring). Reproduced under open access license agreement CC BY-NC-SA 3.0 from Ref. [70]. Copyright 2012 the authors.

**Figure 4 sensors-22-02262-f004:**
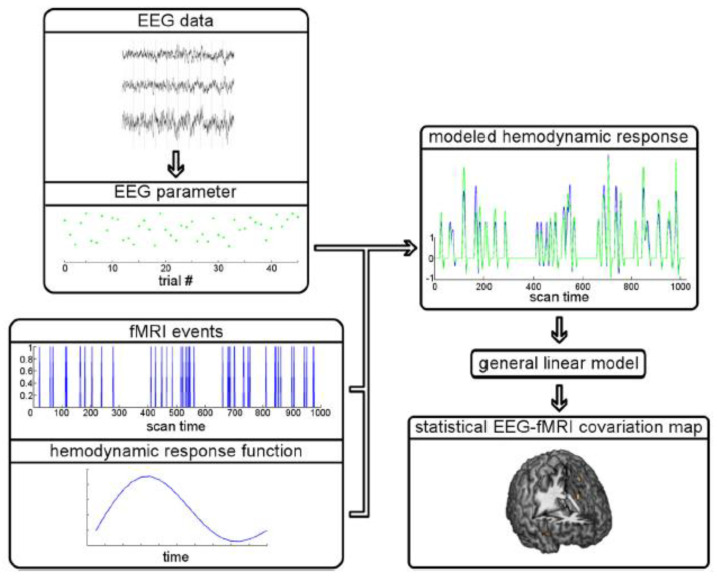
Schematic of EEG-informed fMRI. For an event of interest (e.g., the occurrence of a response error), a parameter value of an EEG feature (e.g., the amplitude of an ERP) is extracted from every trial that includes this event. With respect to fMRI, the onsets of these events during the course of the experiment are known, and the fMRI signal changes caused by haemodynamic responses following the events are mathematically modelled. This is completed via convolution: the multiplication and summation of a vector of zeros and ones representing the event onsets and the haemodynamic response function (mathematically modelled fMRI signal changes following an event). The result is the predicted time course of fMRI signal changes which can then be statistically compared with the observed fMRI signal for each voxel (volume element) of a brain scan. In the case of EEG-informed fMRI, not only is this model determined by event onsets and the haemodynamic response function (blue model prediction), but the expected haemodynamic responses are additionally parameterised using the extracted EEG feature: signal changes following events on trials with a large EEG response are scaled up as compared with trials with smaller EEG responses (green model prediction). Reproduced under open access license agreement CC BY-NC-SA 3.0 from Ref. [70]. Copyright 2012 the authors.

**Figure 5 sensors-22-02262-f005:**
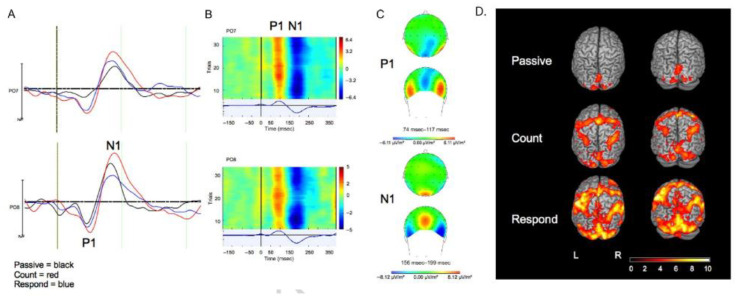
(**A**) Grand average of ERPs at PO7 and PO8 with the P1 and N1 peaks marked. (**B**) Stacked plots of single-trial responses at PO7 and PO8 for a representative subject, illustrating a consistent positivity for the P1 component and a consistent negativity for the N1 component. (**C**) Topography of the P1 (top) and N1 (bottom) components. (**D**) BOLD activation (z scores) in response to target stimuli in the Passive, Count, and Respond conditions (second-level mixed-effects FLAME, *n* = 13, Cluster-corrected threshold Z = 2.0, *p* = 0.05). Reproduced with permission from Ref. [97]. Copyright 2014 MIT Press.

**Figure 6 sensors-22-02262-f006:**
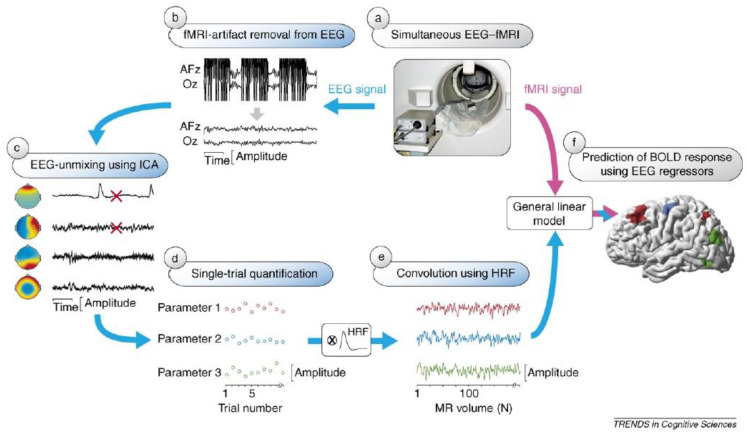
EEG-informed fMRI analysis. EEG (blue arrows) and fMRI (pink arrow) can be recorded simultaneously (**a**) and subsequently, EEG signals are corrected for fMRI artifacts. This is illustrated for two (AFz and Oz) out of a larger number of EEG channels (**b**). ICA applied to the continuous EEG signal returns artifact-related and brain-related component activations and maps; typical artifact-related components are marked with red crosses (**c**). Selected components reflecting brain activity of interest can be used to obtain a measure for each recorded trial (**d**). After convolution with the haemodynamic response function (HRF), the single-trial amplitudes yield EEG regressors (**e**) that parametrically predict the BOLD response (**f**). Reproduced with permission from Ref. [93]. Copyright 2066 Elsevier.

**Figure 7 sensors-22-02262-f007:**
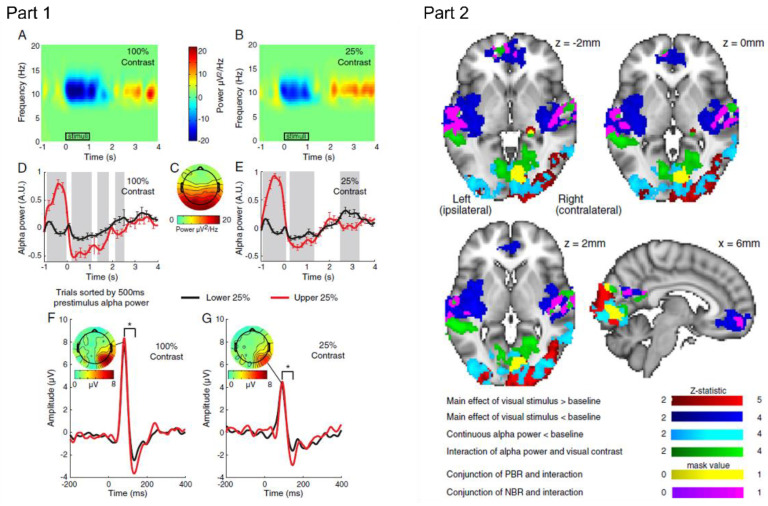
Part 1. Group average time–frequency spectrograms of HC (**A**) and LC (**B**) stimulus induced alpha response with group average bilateral scalp topography of occipital alpha-power (**C**). Black rectangles denote duration of stimulation. Effect of pre-stimulus alpha-power on the HC (**D**) and LC (**E**) alpha-power time course and HC (**F**) and LC (**G**) VEPs. Black and red lines denote lower and upper quartiles of pre-stimulus alpha-power, respectively. Grey shading denotes significant difference (*p* < 0.05) in amplitude between quartiles. Error bars represent standard error in the mean. * denotes significant (*p* < 0.05) difference in VEP P100–N140 amplitude between quartiles. Part 2. Mixed effects group Z-statistic maps (all cluster corrected Z > 2.0, *p* < 0.05) of EEG–fMRI GLM results. Positive (red) and negative (dark blue) BOLD responses to constant amplitude main effect of visual stimulation overlaid with a negative correlation between the BOLD signal and continuous alpha power (light blue). A significant negative interaction between continuous alpha-power and the main effect of the visual stimulus contrast was observed (green) in visual and auditory cortices, precuneus and mPFC. The conjunction of the interaction with visual cortex positive BOLD regions is shown in yellow, and with auditory and DMN negative BOLD regions in purple. Reproduced under open access license agreement CC BY-NC-ND 3.0 from Ref. [100]. Copyright 2013 the authors.

**Figure 8 sensors-22-02262-f008:**
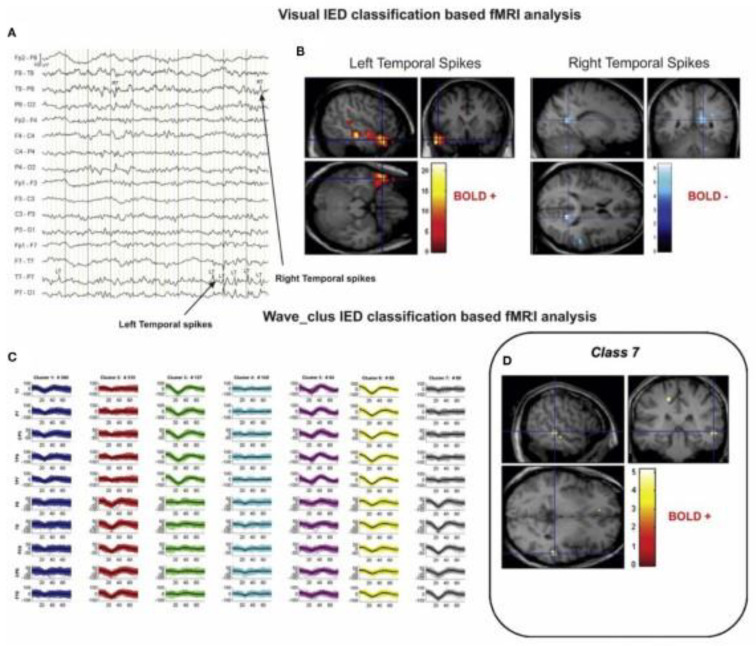
A sample of visual and algorithmic classification of the IEDs. (**A**) The result of visual classification from the bipolar montage (64 channels) of EEG recorded inside the scanner is performed by an expert. (**B**) The results of EEG–fMRI analysis, based on visual-IED labelling. (**C**) The seven classes identified using the algorithmic classification. (**D**) The result of EEG–fMRI analysis, associated with class 7 of the identified IEDs. All the fMRI results are overlaid on the subject’s T1-weighted image (48). Reproduced under open access license agreement CC BY-NC-ND 4.0 from Ref. [106]. Copyright 2021 the authors.

**Figure 9 sensors-22-02262-f009:**
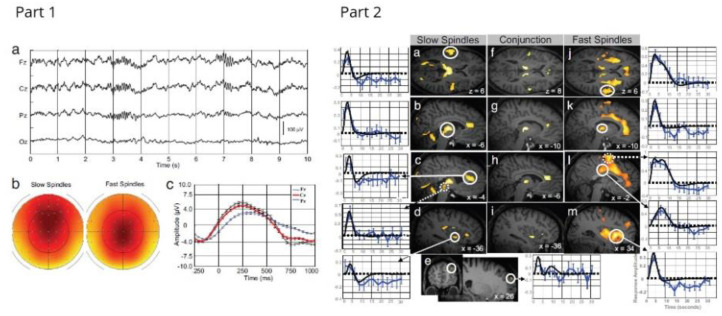
Part 1. EEG characterisation of sleep spindles. (**a**) Example of a typical EEG recording (stage 2 sleep; 0.1 to 70 Hz) after scanner and pulse artifact correction, depicting a fast posterior (Pz, left side) and a slow anterior (Fz, right side) spindle. (**b**) EEG scalp topography of the average spindle band power between 11–13 (left) and 13–15 Hz (right). For display, the average normalised spindle power of all slow and fast sleep spindles (detected on Cz) was computed at each channel and between 11–13 and 13–15 Hz, respectively. Slow sigma power predominates over the frontal central regions whereas fast sigma power is mainly expressed over centro-parietal areas. Nose is upwards, right is rightwards. (**c**) Spindle-triggered average revealing the underlying slow oscillation. EEG data (0.5–4 Hz) were averaged with respect to the onset of all sleep spindles. Spindles start (t 0) on the depolarising phase of the oscillation, which on average are of much smaller amplitude than the classical full blown slow waves of deep, slow-wave sleep. The classical phase lag from frontal (Fz, black) to central (Cz, red) and parietal (Pz, blue) areas and the maximal slow wave amplitude at the frontal recording site are also depicted. Part 2. Main effects of slow and fast sleep spindles. (**a**–**e** Left) fMRI responses to slow spindles displayed over an individual structural image normalised to the Montreal Neurological Institute space (P uncorrected < 0.001). The leftmost panels show peristimulus time histograms (PSTHs) depicting the responses in auditory cortices (circled) (**a**), thalamus (**b**), anterior cingulate (circled) and midbrain tegmentum (dotted) (**c**), anterior insula (**d**), and superior frontal gyrus (**e**). The PSTH (solid blue line; blue error bars reflect the SEM) depicts the mean response across spindles of the corresponding voxel, irrespective of contrast based on a finite impulse response refit. The fitted response is drawn in black. (**f**–**i** Centre) Conjunction analysis of slow and fast sleep spindles. (**j**–**m** Right) fMRI responses to fast spindles (P uncorrected < 0.001). The right most panels show PSTHs depicting the response in superior temporal gyri (**j**), thalami (**k**), mid cingulate cortex (circled) and SMA (dotted) (**l**), and anterior insula (**m**). Reproduced from Ref. [129]. Copyright 2007 National Academy of Sciences.

**Figure 10 sensors-22-02262-f010:**
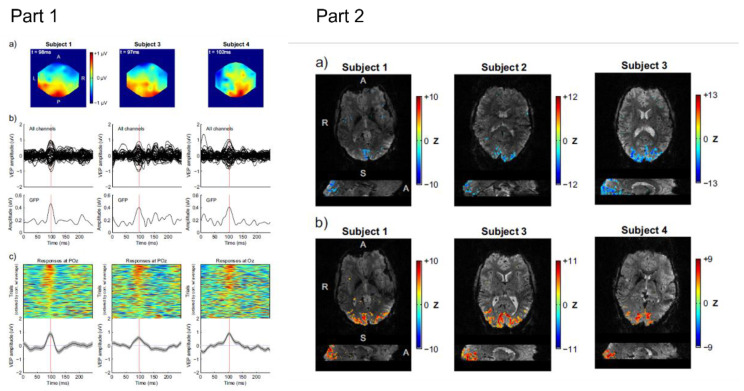
Part 1. Responses to reduced-field reversing checkerboard stimulation in 3 human volunteers, as captured by EEG during simultaneous EEG–fMRI acquisitions. (**a**) Scalp potential maps at the timing of the expected P100 peak. (**b**) Trial-averaged responses of all 63 EEG channels, and the corresponding global field power (GFP) response. (**c**) Average and single-trial responses in a relevant occipital channel (POz or Oz), aligned at the onset of checkerboard reversal (t = 0 ms); single trial responses are ordered from top to bottom according to their correlation with the average response; only the 200 best trials (out of 312) are displayed. All results shown are derived from the reconstructed EEG datasets following ICA decomposition and source selection. Part 2. BOLD responses to (**a**) an eyes-open/eyes-closed task and (**b**) a reduced-field reversing checkerboard stimulation run, expressed as Z-score statistical maps. (**a**) Negative values reflect negative BOLD signal changes during eyes-closed periods; maps were thresholded at Z = −3.5. (**b**) Positive values reflect positive signal changes during checkerboard stimulation periods; maps were thresholded at Z = +3.5. Colour bar ranges were manually restricted for clearer visualisation. Reproduced with permission from Ref. [22]. Copyright 2015 Elsevier.

**Figure 11 sensors-22-02262-f011:**
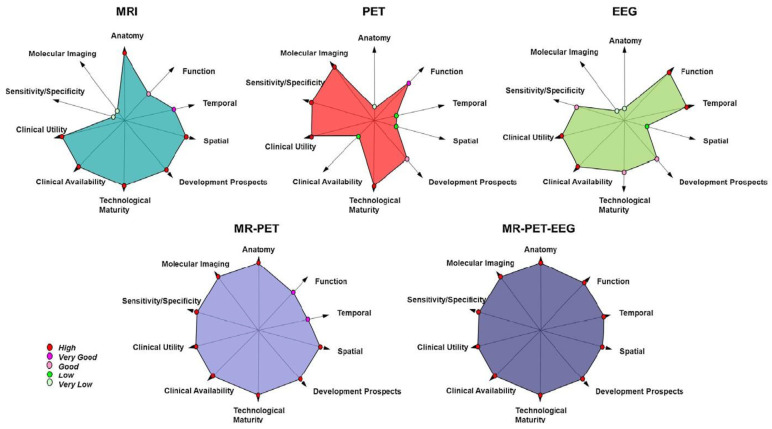
Fingerprint diagrams giving an overview of the strengths of MRI, PET, EEG, hybrid MR–PET, and hybrid MR–PET–EEG. Starting at the origin, the further one traverses along a given axis, the better that particular attribute is fulfilled. MRI can provide exquisite spatial resolution and the technology is widely available; however, MRI is not strong in the area of molecular imaging and its specificity is also somewhat limited. PET on the other hand, has poorer spatial and temporal resolution than MRI but it is extremely specific—an attribute conferred upon it by the choice of radiolabelled tracer—and is also very sensitive. Both MRI and PET have a poor temporal resolution regarding the mapping of brain function, for example. In a hybrid scanner capable of simultaneous measurement of all three datasets, all the chosen attributes are fulfilled in entirety. Reproduced with permission from Ref. [160]. Copyright 2013 Elsevier.

## Data Availability

Not applicable.

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
