# Peer review of "Simultaneous EEG-fMRI: What Have We Learned and What Does the Future Hold?"

_sensors, 2022, doi:10.3390/s22062262_

Round 1

Reviewer 1 Report

This is an important review and summary of the applications and future of combined EEG and fMRI.

Here is a suggestion to help improve the manuscript.

In section 4. Applications of EEG-fMRI

I recommend that example EEG and fMRI figure for each type of application be presented.

For example

4.2. Event-related responses

4.3. EEG frequency content

4.4. Epilepsy

4.5. Sleep

5.2. Ultra-high field EEG-fMRI

In each of these sections, it would be helpful if the authors could show example figure of real data of both EEG and fMRI.

Author Response

I would like to thank the reviewer for their encouraging comments and for the suggestion to include figures showing EEG and fMRI data for each application. I have included a figure in all sections suggested by the reviewer.

4.2. Event-related responses: Figure 5 has been added

4.3. EEG frequency content: Figure 7 has been added

4.4. Epilepsy: Figure 8 has been added

4.5. Sleep: Figure 9 has been added

5.2. Ultra-high field EEG-fMRI: Figure 10 has been added

Reviewer 2 Report

Simultaneous EEG-fMRI is one of the most powerful tools available for the non-invasive study of human brain function. In the current review the authors in details present the methodology, advantages, disadvantages and safety of each technique separately in order to highlight the importance of the concurrent EEG-fMRI in research and the development of clinical interventions mainly in the field of psychiatry and neurology. The study fits in the scope of the journal, is well organized and the data are clearly presented;  the figures are very helpful especially for the non expert. A table showing the exciting research possibilities and clinical applications of EEG-fMRI and in combination with other techniques i.e brain stimulation techniques, EEG-fMRI-PET, would be useful

Author Response

I would like to thank the reviewer for their encouraging comments.

I have added the suggested table (Table 1). However, I’m not completely sure that I understood what information the reviewer would like to see in the table. The combination of EEG-fMRI with other methods is a relatively small body of literature and while I completely agree that this is an interesting topic with some exciting potential applications, some of the multi-modal studies (e.g. brain stimulation) are in healthy populations, and only speculate on potential clinical applications. I have split the table into studies that investigate a clinical population and those that only suggest a clinical application. I hope this fits with the reviewer’s expectations. I am also happy to consider not including the table if speculating about clinical applications is not considered appropriate.